# Zero-shot Node Classification with Graph Contrastive Embedding Network

**Wei Ju**[1]                                    *juwei@pku.edu.cn*
**Yifang Qin**[1]                               *qinyifang@pku.edu.cn*
**Siyu Yi**[2]                                  *siyuyi@mail.nankai.edu.cn*
**Zhengyang Mao**[1]                        *zhengyang.mao@stu.pku.edu.cn*
**Kangjie Zheng**[1]                           *kangjie.zheng@gmail.com*
**Luchen Liu**[1]                               *liuluchen@pku.edu.cn*
**Xiao Luo**[3,*]                               *xiaoluo@cs.ucla.edu*
**Ming Zhang**[1,*]                            *mzhang_cs@pku.edu.cn*

[1]*School of Computer Science, National Key Laboratory for Multimedia Information Processing, Peking University*
[2]*School of Statistics and Data Science, Nankai University*
[3]*Department of Computer Science, University of California, Los Angeles*

**Reviewed on OpenReview:** *https://openreview.net/forum?id=8wGXnjRLSy*

## Abstract

This paper studies zero-shot node classification, which aims to predict new classes (i.e., unseen classes) of nodes in a graph. This problem is challenging yet promising in a variety of real-world applications such as social analysis and bioinformatics. The key idea of zero-shot node classification is to enable the knowledge transfer of nodes from training classes to unseen classes. However, existing methods typically ignore the dependencies between nodes and classes, and fail to be organically integrated in a united way. In this paper, we present a novel framework called the Graph Contrastive Embedding Network (GraphCEN) for zero-shot node classification. Specifically, GraphCEN first constructs an affinity graph to model the relations between the classes. Then the node- and class-level contrastive learning (CL) are proposed to jointly learn node embeddings and class assignments in an end-to-end manner. The two-level CL can be optimized to mutually enhance each other. Extensive experiments indicate that our GraphCEN significantly outperforms the state-of-the-art approaches on multiple challenging benchmark datasets.

## 1 Introduction

Zero-shot Learning (ZSL) is an important problem in machine learning, which has been extensively studied in computer vision. It aims at recognizing samples from novel classes that have never appeared in the training data, whose potential has attracted a lot of research interests in a wide range of tasks, such as image classification (Wang et al., 2021a; Li et al., 2022), object recognition (Zablocki et al., 2019; Han et al., 2020) and knowledge graph completion (Geng et al., 2022; Nayak & Bach, 2022). To effectively classify those newly emerging classes, existing ZSL approaches typically transfer knowledge from the classes that have training samples (i.e., seen classes) to these unseen classes without requiring any explicitly labeled data for these unseen classes. This can be achieved by the guidance of some auxiliary semantic information (e.g., category attributes or word embeddings), which usually establishes the mathematical relationship between the semantic space and the embedding space.

---

*Corresponding authors.

Recently, due to the unprecedented success of deep learning, deep neural networks have extended their excellent representation learning capability from the field of vision to graph-structured data. With the advancement of graph neural networks (GNNs) (Kipf & Welling, 2017; Ju et al., 2023a; Nagarajan & Raghunathan, 2023; Ma et al., 2023; Chen et al., 2023; Luo et al., 2023b), node classification, as one of the most important problems in graph data analysis, has been widely investigated by various types of GNNs. The basic idea of node classification is to predict the unlabeled nodes with only a small number of labeled nodes on the graph, and it is usually assumed that all classes are covered by the classes of labeled nodes. Nevertheless, the graph typically evolves in dynamic and open environments. When newly emerging classes come, previously trained GNNs fail to be discriminative to unseen classes, and massive nodes from novel classes need to be collected and labeled. However, it is costly and laborious to annotate enough samples and retrain the model for all the newly emerging classes. It thus naturally raises a meaningful question: *can we recognize the nodes from novel classes that have never appeared?*

Toward this end, zero-shot node classification on graphs has become a promising approach to achieve the goal. Despite the encouraging achievements of previous GNN methods via message-passing mechanism (Gilmer et al., 2017), they still suffer from three key limitations: (i) **Under-explored task on graphs**. Zero-shot node classification is a newly emerging task that remains largely unexplored, while existing GNN methods for traditional node classification cannot handle this problem. This is because traditional GNNs are typically trained on a set of labeled nodes and fail to generalize to unseen classes. (ii) **Inability to explicitly model the dependencies between nodes and classes**. A vast majority of existing GNNs typically learn a mapping function from features to labels to implicitly capture the relationship between nodes and classes, i.e., class information is only incorporated by computing cross-entropy loss with the output embedding vectors. In other words, class labels fail to be explicitly used to enhance information propagation through the model, which is often sub-optimal. Moreover, explicitly modeling such dependencies is crucial for zero-shot node classification, since when facing newly emerging classes, we can leverage these dependencies to effectively align the mapping relationship between the node features and the new class labels, thereby enhancing the model's generalization ability. (iii) **Fail to jointly learn node and class embeddings in a united way**. Existing approaches mainly concentrate on learning effective node embeddings for class assignments, while failing to reward the learning of node embeddings from the perspective of classes in turn. As such, we are looking for an approach tailored to zero-shot node classification that can model the dependencies between nodes and classes, and meanwhile jointly learn node and class embeddings in a united way.

To address the above issues, this work proposes a novel framework called the Graph Contrastive Embedding Network (GraphCEN) for zero-shot node classification. The key idea of GraphCEN is to exploit the multi-granularity information to provide sufficient evidence on establishing the relationship between the training classes and newly emerging classes to link each other. To achieve this goal, GraphCEN first constructs an affinity graph of classes to incorporate category semantic knowledge, we then leverage GNNs to effectively integrate the node feature and the class semantics to learn a joint information matrix. Grounded in this, we introduce two-level contrastive learning (CL) based on the joint information matrix, i.e., a node-level CL and a class-level CL, respectively. On the one hand, node-level CL is conducted in the row space of the information matrix to learn node embeddings for effective class assignments. On the other hand, class-level CL is achieved in the column space of the information matrix to capture compact class embeddings encouraging class-level consistency. Thus the representation learning and class assignment can be jointly optimized to collaborate with each other. By incorporating this multi-granularity information, our experiments on multiple real-world datasets prove that our GraphCEN can substantially improve the performance against existing state-of-the-art approaches. To summarize, the main contributions of this work are as follows:

- **General Aspects:** We explore a challenging yet promising problem: zero-shot node classification on graphs, which is under-explored in graph machine learning and data mining.

- **Novel Methodologies:** We present a novel framework to explore node- and class-level contrastive learning based on the joint information matrix. Node-level CL aims to learn effective node embeddings, while class-level CL captures discriminative class embeddings.

- **Multifaceted Experiments:** Experimental results on three benchmark datasets demonstrate that our proposed GraphCEN outperforms the state-of-the-art approaches.

## 2    Problem Definition & Preliminary

We first formalize the definition of the graph and define the core problem of our paper: zero-shot node classification. Subsequently, we introduce several types of data augmentation.

**Definition 1: Graph.** Let $\mathcal{G} = (\mathcal{V}, \mathcal{E})$ denote a graph, where $\mathcal{V} = \{v_1, \cdots, v_N\}$ represents the set of nodes, in which $N$ is the number of nodes. $\mathcal{E} \subseteq \mathcal{V} \times \mathcal{V}$ represents the set of edges. We denote the feature matrix as $\mathbf{X} \in \mathbb{R}^{N \times d}$, where $\mathbf{x}_i \in \mathbb{R}^d$ is the feature of the node $v_i$, and $d$ is the dimension of features. $\mathbf{A} \in \{0,1\}^{N \times N}$ describes the adjacency matrix, where $a_{ij} = 1$ if $(v_i, v_j) \in \mathcal{E}$ otherwise $a_{ij} = 0$.

**Definition 2: Zero-shot Node Classification.** Denote by $\mathcal{C} = \{c_1, \ldots, c_{|\mathcal{C}|}\}$ the whole class set, which can be divided into the seen class set $\mathcal{C}_s = \{c_1, \ldots, c_{|\mathcal{C}_s|}\}$ and the unseen class set $\mathcal{C}_u = \{c_{|\mathcal{C}_s|+1}, \ldots, c_{|\mathcal{C}|}\}$, satisfying $\mathcal{C}_s \cup \mathcal{C}_u = \mathcal{C}$, and $\mathcal{C}_s \cap \mathcal{C}_u = \emptyset$. Let $\mathbf{S} = (\mathbf{s}_1, \ldots, \mathbf{s}_{|\mathcal{C}|})^\top$ denote the semantic description matrix of all classes, each row of which corresponds to the description of a class. Class semantic descriptions (CSDs) have been detailedly studied by (Wang et al., 2021b). For zero-shot node classification, assuming all the labeled nodes are from seen classes $\mathcal{C}_s$, based on the CSDs, the goal of zero-shot node classification is to classify the unlabeled nodes whose class set is $\mathcal{C}_u$.

**Graph Augmentations.** For contrastive learning, data augmentation is crucial that produces novel rational data by applying certain transformations without altering the semantics. In particular, in our experiments, we introduce two types of graph transformations via augmenting topological information of the graphs respectively, formalized as:

- **Edge Dropping**. It randomly deletes a certain ratio of edges in the graph to perturb the edge connectivity pattern where probability follows a default i.i.d. uniform distribution. The underlying assumption is that the semantic information of the graph has a certain robustness to the edge connectivity pattern variances.

- **Graph Diffusion**. It leverages diffusion (Klicpera et al., 2019) to provide a congruent view of the original graph, which contributes to offering global information. Here we adopt the Personalize PageRank (PPR) kernel to characterize graph diffusion as

$$\mathbf{A}' = \alpha \left( \mathbf{I} - (1 - \alpha)\mathbf{D}^{-1/2}(\mathbf{A} + \mathbf{I})\mathbf{D}^{-1/2} \right)^{-1}, \tag{1}$$

  where $\alpha$ is the teleport probability which is set to 0.2 as default. $\mathbf{A}$, $\mathbf{D}$, $\mathbf{I}$ represent the adjacency matrix, the degree matrix and the identity matrix, respectively. We can replace $\mathbf{A}$ with $\mathbf{A}'$ as an augmented view, which can capture long-range neighborhood information.

## 3    Methodology

### 3.1    Overview

This paper presents a novel framework GraphCEN for zero-shot node classification as shown in Figure 1. At a high level, GraphCEN aims to leverage the multi-granularity information to jointly learn node embeddings and class assignments in a united fashion. Specifically, GraphCEN first constructs a class affinity graph to incorporate category semantic knowledge and capture the relations between all classes, and then the joint information matrix is obtained by combining the node feature and the class semantics via GNNs. Further, the node- and class-level contrastive learning are respectively conducted in the row and column spaces of the joint information matrix to learn node and class embeddings for effective class assignments. The two steps can be jointly trained to collaborate with each other. In the following, we will elaborate on each component of our proposed framework GraphCEN in turn.

### 3.2    Class Affinity Graph Construction

A key to zero-shot learning (ZSL) is the acquisition of the class semantic descriptions (CSDs), which serve as the auxiliary data to represent the semantics of the classes. However, CSDs of the categories are indepen-

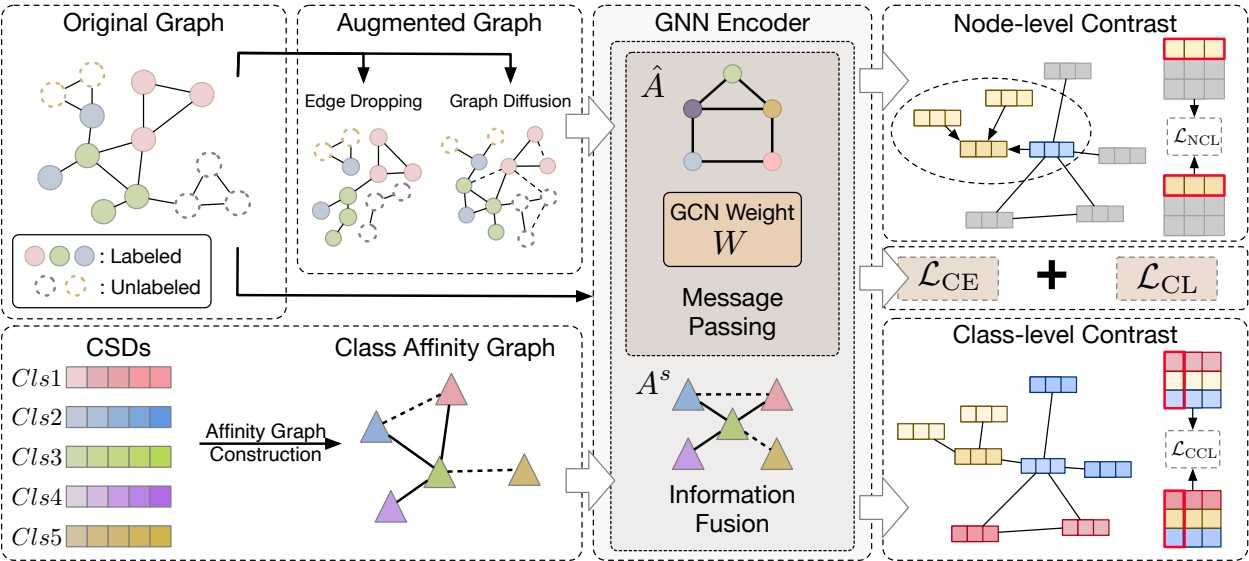

Figure 1: Illustration of the proposed framework GraphCEN.

dently presented based on their own text descriptions or word embedding. In other words, the relationship among different classes is ignored, which is crucial for transferring class semantic knowledge from seen classes to unseen classes. As such, how to capture the relationship between different classes, is a key factor for enhancing the knowledge transfer among all classes and generalization ability under the zero-shot setting.

Therefore, we propose to construct a class affinity graph to capture the relationship among different classes, such that the semantic information of categories can communicate with each other to allow for semantic knowledge transfer, thus benefiting the better classification for unseen classes.

Specifically, based on the CSDs $\mathbf{S} = (\mathbf{s}_1, \ldots, \mathbf{s}_{|\mathcal{C}|})^\top$, we construct a class affinity graph $G_A = (\mathcal{V}_A, \mathcal{E}_A)$, where $\mathcal{V}_A = \{c_1, \cdots, c_{|\mathcal{C}|}\} = \mathcal{C}$ is the node set consisting of all classes and $\mathcal{E}_A$ is the edge set. $\mathcal{E}_A$ is described by the class adjacency matrix $\mathbf{A}^{\mathbf{S}} \in \mathbb{R}^{|\mathcal{C}| \times |\mathcal{C}|}$, each of whose entry has the form

$$\mathbf{A}_{ij}^{\mathbf{S}} = \begin{cases} e^{\mathbf{s}_i^\top \mathbf{s}_j}, & \text{if } c_j \in \mathcal{N}(c_i) \\ 0, & \text{otherwise} \end{cases}. \tag{2}$$

The neighborhood set $\mathcal{N}(c_i)$ of class $c_i$ is determined by the $k$-nearest neighbors of $\mathbf{s}_i$ in the CSDs.

In this way, the class affinity graph $G_A = (\mathcal{V}_A, \mathcal{E}_A)$ overcomes the weakness of the independence of CSDs, and the relationship among all classes can be well captured to provide abundant prior knowledge, better serving the zero-shot node classification on graphs. Based on the above class affinity graph, we can leverage a well-designed graph-based encoder to effectively exploit and propagate the semantics of different classes, promoting knowledge transfer from seen classes to unseen classes, which is shown in the following section.

### 3.3 Information Fusion on Graphs

Existing graph neural networks (GNNs) (Kipf & Welling, 2017; Veličković et al., 2017; Ju et al., 2022; Luo et al., 2023a) typically model the mapping function from nodes to classes through message-passing mechanism (Gilmer et al., 2017), which encode the structural and attributive information into node embeddings. The key idea is to learn the embedding of each node via aggregating its own embeddings and the ones of its neighbors' along edges. The whole process can be formalized as:

$$\mathbf{V} = \text{ReLU}(\hat{\mathbf{A}}\mathbf{X}\mathbf{W}), \tag{3}$$

where $\mathbf{V} \in \mathbb{R}^{N \times |\mathcal{C}|}$ denotes the embedding matrix of nodes. $\hat{\mathbf{A}} = \tilde{\mathbf{D}}^{-\frac{1}{2}} \tilde{\mathbf{A}} \tilde{\mathbf{D}}^{-\frac{1}{2}}$ where $\tilde{\mathbf{A}} = \mathbf{A} + \mathbf{I}$ and $\tilde{\mathbf{D}}$ is the degree matrix of $\tilde{\mathbf{A}}$. $\mathbf{W} \in \mathbb{R}^{d \times |\mathcal{C}|}$ is the trainable weight matrix.

However, a vast majority of these GNNs implicitly capture the relationship between nodes and classes via the mapping function, which often leads to sub-optimal solutions. Hence, we argue that explicitly incorporating the relations between the classes and modeling the dependencies between nodes and classes is crucial for effective class assignments. To that effect, on the strength of the constructed class affinity graph $\mathbf{A^S}$ above, we integrate the node feature and the class semantics to learn a joint information matrix defined as:

$$\mathbf{G} = \text{ReLU}(\hat{\mathbf{A}}\mathbf{X}\mathbf{W})\mathbf{A^S}, \tag{4}$$

In this way, both the semantics of each node and each class can be efficiently propagated along their respective edges, which contributes to learning the joint information matrix of the nodes and the classes. Moreover, $\mathbf{G} \in \mathbb{R}^{N \times |\mathcal{C}|}$ is trainable and can be updated for sufficiently combining both sides to simultaneously keep the relations between the nodes and between the classes in a united way.

### 3.4 Two-level Contrastive Learning

Grounded in the joint information matrix of the nodes and the classes defined above, we found a key observation that the rows and columns of the joint information matrix could be respectively regarded as the embeddings of nodes and classes. By jointly optimizing the node- and class-level embeddings, we might learn effective representations for class assignments, which result in better learning performance.

Inspired by the remarkable success of contrastive learning (CL), which has demonstrated the strong ability to learn discriminative embeddings from the data itself, we marry this technique into our model for better representation learning. The underlying concept of CL is to explicitly compare pairs of sample embeddings to push away embeddings from different samples while pulling together those from augmentations of the same sample. Given this, we propose two-level contrastive learning (i.e., node-level and class-level) based on our joint information matrix to jointly optimize node and class embeddings in a united way.

#### 3.4.1 Node-level Contrastive Learning

As the rows of the joint information matrix (i.e., node embeddings of the graph) could be treated as the class assignment probabilities since the dimensionality of the rows equals the number of classes. As such, effective embedding is beneficial for generating a more confident class assignment.

Technically, we first randomly choose one of the two graph augmentations in Section 2 to construct node positive/negative pairs. After encoding them in our well-designed GNN of Section 3.3, an additional node projection head (i.e., multi-layer perceptron, MLP) is adopted to map the joint information matrices to the space where contrastive loss is applied. Then we obtain the node-level augmented joint embedding matrices $\mathbf{H}^a \in \mathbb{R}^{N \times d'}$ and $\mathbf{H}^b \in \mathbb{R}^{N \times d'}$. Formally, $\mathbf{H}^a = \Psi(\text{ReLU}(\hat{\mathbf{A}}^a \mathbf{X}\mathbf{W})\mathbf{A^S})$ and $\mathbf{H}^b = \Psi(\text{ReLU}(\hat{\mathbf{A}}^b \mathbf{X}\mathbf{W})\mathbf{A^S})$, where $\Psi(\cdot)$ is the projection function with parameter matrix $\mathbf{\Theta}_1$, $\hat{\mathbf{A}}^x = \tilde{\mathbf{D}}^{-\frac{1}{2}} \text{Aug}_x(\mathbf{A})\tilde{\mathbf{D}}^{-\frac{1}{2}}$, $\text{Aug}_x$ is the augmentation operation, and $\tilde{\mathbf{D}}$ is the corresponding degree matrix of $\text{Aug}_x(\mathbf{A})$ for $x \in \{a, b\}$. Therefore, given $2B$ augmented node embeddings $\{\mathbf{h}_1^a, \ldots, \mathbf{h}_B^a, \mathbf{h}_1^b, \ldots, \mathbf{h}_B^b\}$ from the perspective of the rows, where $B$ is the size of the mini-batch, we expect that the two augmented samples from the same node should be pulled closer, while different nodes should be pushed away. In other words, there are $2B - 1$ pairs for a target node $h_i^a$, then we select its corresponding augmented node $h_i^b$ to construct a positive pair $\{\mathbf{h}_i^a, \mathbf{h}_i^b\}$ and leave other $2B - 2$ pairs to form negative pairs. In this way, the training objective of node-level contrastive learning for a given node $h_i^a$ is formulated as:

$$\mathcal{L}_i^a = -\log \frac{e^{\text{sim}(\mathbf{h}_i^a, \mathbf{h}_i^b)/\tau}}{\sum_{j=1}^B \left(e^{\text{sim}(\mathbf{h}_i^a, \mathbf{h}_j^a)/\tau} + e^{\text{sim}(\mathbf{h}_i^a, \mathbf{h}_j^b)/\tau}\right)}, \tag{5}$$

where $\tau$ denotes the temperature parameter and $\text{sim}(\mathbf{h}_1, \mathbf{h}_2)$ is the cosine similarity $\frac{\mathbf{h}_1^\top \mathbf{h}_2}{\|\mathbf{h}_1\| \cdot \|\mathbf{h}_1\|}$.

Also, two augmented nodes are mirrored and can be switched, hence the total node-level contrastive learning loss is computed over every augmented sample defined as:

$$\mathcal{L}_{\text{NCL}} = \frac{1}{2B} \sum_{i=1}^B (\mathcal{L}_i^a + \mathcal{L}_i^b). \tag{6}$$

As such, minimizing $\mathcal{L}_{NCL}$ can achieve the goal that the learned node embeddings have clearer boundaries to distinguish between different sample pairs, such that learned discriminative embeddings are beneficial for effective class assignments.

### 3.4.2 Class-level Contrastive Learning

In addition to the node embeddings derived from the rows of the joint information matrix, correspondingly, with a dimensionality of $|\mathcal{C}|$, the columns of the matrix can be viewed as class embeddings. In other words, the columns can be interpreted as the distribution of the classes over nodes.

Moreover, the key to zero-shot learning is establishing the mathematical relationship between all classes and transferring knowledge from the seen classes to the unseen classes. Inspired by this motivation, conducting CL on class embeddings can make the different category information well capture the inherent characteristics.

Technically, similar to Section 3.4.1, we obtain the class-level augmented joint embedding matrices $\mathbf{Z}^a \in \mathbb{R}^{N \times |\mathcal{C}|}$ and $\mathbf{Z}^b \in \mathbb{R}^{N \times |\mathcal{C}|}$ by encoding the augmented graphs in our well-designed GNN followed by a class projection head (i.e., MLP) with parameter matrix $\mathbf{\Theta}_2$. Therefore, $\mathbf{Z}^a_{i,c}$ can be regarded as the probability of node $i$ being assigned to class $c$. Given $2|\mathcal{C}|$ augmented node embeddings $\{\mathbf{z}^a_1, ..., \mathbf{z}^a_{|\mathcal{C}|}, \mathbf{z}^b_1, ..., \mathbf{z}^b_{|\mathcal{C}|}\}$ from the perspective of the columns, where each embedding $\mathbf{z}^a_i$ denotes the $i$-the column of $\mathbf{Z}^a$, namely, the embedding of class $i$ under the first graph augmentation. Similarly, we encourage the positive pairs to be similar in the embedding space while pushing away the negative pairs. Thus we can leverage the idea of CL to define the training objective of class-level contrastive learning as:

$$\hat{\mathcal{L}}^a_i = -\log \frac{e^{\mathrm{sim}(\mathbf{z}^a_i, \mathbf{z}^b_i)/\tau}}{\sum_{j=1}^{|\mathcal{C}|} \left( e^{\mathrm{sim}(\mathbf{z}^a_i, \mathbf{z}^a_j)/\tau} + e^{\mathrm{sim}(\mathbf{z}^a_i, \mathbf{z}^b_j)/\tau} \right)}, \tag{7}$$

By traversing all classes, the total class-level contrastive learning loss is computed as:

$$\mathcal{L}_{\mathrm{CCL}} = \frac{1}{2|\mathcal{C}|} \sum_{i=1}^{|\mathcal{C}|} (\hat{\mathcal{L}}^a_i + \hat{\mathcal{L}}^b_i) - H(\mathbf{Z}), \tag{8}$$

where $H(\cdot)$ represents the entropy function to prevent collapsing into trivial outputs of the same class. Mathematically, $H(\mathbf{Z}) = \sum_{i=1}^{|\mathcal{C}|} -\mathbf{p}^a_i \log \mathbf{p}^a_i - \mathbf{p}^b_i \log \mathbf{p}^b_i$ where $\mathbf{p}^x_i = \sum_{t=1}^{N} \mathbf{Z}^x_{ti} / ||\mathbf{Z}^x||$, $x \in \{a, b\}$ is the normalized probability vector. Here, compared with learning discriminative class representations through contrastive learning to implicitly prevent model collapse, we directly introduce the entropy function to encourage each column of the augmented joint embedding matrix to be close to a uniform distribution, explicitly avoiding the trivial solution that most nodes are assigned to the same class (Hu et al., 2017).

In this way, class-level contrastive learning can reward the optimization of node-level contrastive learning, and guide the learning of node embeddings for effective class assignments.

### 3.5 Training and Optimization

To jointly integrate the two-level contrastive learning, we combine the two contrastive losses to collaborate with each other. And the training objective can be written as:

$$\mathcal{L}_{\mathrm{CL}} = \mathcal{L}_{\mathrm{NCL}} + \mathcal{L}_{\mathrm{CCL}}. \tag{9}$$

Moreover, we can also incorporate supervision signals from the labeled nodes from seen classes $\mathcal{C}_s$, where the cross-entropy loss function can be defined as:

$$\mathcal{L}_{\mathrm{CE}} = -\sum_{i=1}^{L} \sum_{j=1}^{|\mathcal{C}_s|} \mathbf{Y}^{\mathrm{true}}_{ij} \ln \mathbf{Y}_{ij}, \tag{10}$$

where $\mathbf{Y}_{ij} = \mathrm{softmax}(\hat{\mathbf{A}}\mathbf{X}\mathbf{W})$ denotes the predicting probability of the $i$-th node belonging to class $j$, $L$ is the number of the labeled nodes.

---

**Algorithm 1:** Optimization Algorithm of GraphCEN

---

**Input** : Graph data $\mathcal{G} = (\mathcal{V}, \mathcal{E}, \mathbf{X}, \mathbf{A})$, classes semantic descriptions matrix $\mathbf{S}$, total classe set $\mathcal{C}$, seen classes $\mathcal{C}_s$, trainable weight matrix $\mathbf{W}$ in GNN, parameter matrix $\mathbf{\Theta} = (\mathbf{\Theta}_1^\top, \mathbf{\Theta}_2^\top)^\top$ in two projection heads, temperature parameter $\tau$ and contrastive parameter $\beta$

**Output:** Class assignments for unlabeled nodes from unseen class $\mathcal{C}_u$

Construct the class affinity graph $G_A$ by Eq. 2;

Initializing trainable weight matrix $\mathbf{W}$ and parameter matrix $\mathbf{\Theta}$;

**while** not done **do**

    Compute the joint information matrix $\mathbf{G}$ by Eq. 4;

    Sample a mini-batch of nodes from $\mathcal{G}$;

    Sample one augmentation from Section 2;

    Compute node-level contrastive loss $\mathcal{L}_{\text{NCL}}$ by Eq. 6;

    Compute class-level contrastive loss $\mathcal{L}_{\text{CCL}}$ by Eq. 8;

    Compute the supervised loss $\mathcal{L}_{\text{CE}}$ by Eq. 10;

    Update parameters $\mathbf{W}$ and $\mathbf{\Theta}$ by gradient descent to minimize $\mathcal{L}$ by Eq. 11

**end**

---

Finally, by combining supervised loss $\mathcal{L}_{\text{CE}}$ with two-level contrastive loss $\mathcal{L}_{\text{CL}}$, the overall loss function of our GraphCEN can be calculated as:

$$\mathcal{L} = \mathcal{L}_{\text{CE}} + \beta \cdot \mathcal{L}_{\text{CL}}, \tag{11}$$

where $\beta$ is the tuning parameter controlling the magnitude of supervised loss and two-level contrastive loss. We summarize the optimization algorithm for our GraphCEN in Algorithm 1.

### 3.6 Computational Complexity Analysis

Suppose that $N$ is the number of nodes, $|\mathcal{E}|$ is the number of edges, $B$ is the batch size, $d$ is the dimension of original node features, and $d'$ is the dimension of the node-level joint embedding matrices after a projection head. We compute the joint information matrix in $O(|\mathcal{E}|d + Nd|\mathcal{C}| + N|\mathcal{C}|^2)$ based on a sparse adjacency matrix and the complexity of node- and class-level projection heads is $O(Ndd' + Nd|\mathcal{C}|)$. Moreover, the complexities of calculating node- and class-level contrastive losses for each batch are $O(B^2d')$ and $O(|\mathcal{C}|^2B)$, respectively. Therefore, the total computational complexity is $O(|\mathcal{E}|d + Nd|\mathcal{C}| + N|\mathcal{C}|^2) + O(Ndd' + Nd|\mathcal{C}|) + O(|\frac{N}{B}|(B^2d' + |\mathcal{C}|^2B)) = O(|\mathcal{E}|d + N(|\mathcal{C}|d + (B+d)d' + 2|\mathcal{C}|^2))$.

## 4 Experiments

### 4.1 Experimental Setup

**Datasets.** For comprehensive comparisons, we conduct extensive experiments on three real-world citation datasets, which are Cora (McCallum et al., 2000), Citeseer (Giles et al., 1998), C-M10M (Pan et al., 2016) and a large-scale dataset ogbn-arxiv (Wang et al., 2020).

In these four datasets, nodes represent different publications, and edges represent the citation relationship between the linked two publications. The specific class labels of the datasets are displayed in Table 1. For the zero-shot node classification, we follow the same seen/unseen class split settings introduced in (Wang et al., 2021b) to make a fair comparison as shown in Table 1. Besides, we adopt two kinds of CSDs, i.e., TEXT-CSDs (default) and LABEL-CSDs, generated by Bert-Tiny as auxiliary data following (Wang et al., 2021b), which can provide the semantic information of different labels. We will further compare the differences between these CSDs in Section 4.5.

**Baseline Methods.** On the one hand, we compare our GraphCEN with various state-of-the-art zero-shot learning methods including DAP and its variant DAP(CNN) (Lampert et al., 2013), ESZSL (Romera-Paredes & Torr, 2015), ZS-GCN and its variant ZS-GCN(CNN) (Wang et al., 2018), WDVSc (Wan et al., 2019), and Hyperbolic-ZSL (Liu et al., 2020), which are primarily proposed for vision domains. On the other hand,

Table 1: The data split and class information of three citation datasets.

| Dataset | Classes | Class Split I [Train/Val/Test] | Class Split II [Train/Val/Test] | Class Labels |
|---|---|---|---|---|
| Cora | 7 | [3/0/4] | [2/2/3] | Neural Network, Rule Learning, Reinforcement Learning, Probabilistic Methods, Theory, Genetic Algorithms, Cased based |
| Citeseer | 6 | [2/0/4] | [2/2/2] | Agent, Information Retrieval, Database, Human Computer Interaction, Artificial Intelligence, Machine Learning |
| C-M10M | 6 | [3/0/3] | [2/2/2] | Biology, Computer Science, Finacial Economics, Industrial Engineering, Physics, Social Science |
| ogbn-arxiv | 40 | [20/0/20] | [13/13/14] | cs.AI, cs.LG, cs.OS, ... |

two recent methods designed for zero-shot node classification, DGPN (Wang et al., 2021b) and DBiGCN (Yue et al., 2022), are also considered for comparison. In addition, we adopt the RandomGuess as the naive baseline, which guesses the unseen labels for the unlabeled nodes randomly.

**Implementation Details.** For our GraphCEN, we adopt the grid search for the parameters under the class split I, and determine the optimal hyper-parameters by the validation classes under the class split II. The space of hyper-parameters is carefully selected as follows: the learning rate $\in \{0.0005, 0.001, 0.005, 0.01\}$, the number of hidden units $\in \{32, 64, 128, 256\}$, the temperature parameter $\tau \in \{0.05, 0.1, 0.2, 0.5, 1.0, 2.0, 5.0, 10.0\}$, the contrastive parameter $\beta \in \{0.05, 0.1, 0.2, 0.5, 1.0, 2.0, 5.0, 10.0\}$. Besides, the framework is trained using Adam optimizer (Kingma & Ba, 2014). In our experiment, we use accuracy as the common evaluation metric to evaluate the performance.

## 4.2 Experimental Results

In this section, we evaluate the performance of all the algorithms for zero-shot node classification. The results are summarized in Table 2. According to the quantitative results, we have the following observations:

- Overall, from the results, it can be observed that our framework GraphCEN achieves the best performance against other strong baselines on all three datasets under different class split settings. In particular, GraphCEN outperforms the closest competitor 7.43% on Cora under the class split I and 7.12% on C-M10M under the class split II, which demonstrates the excellent capability of our framework for zero-shot node classification.

- Traditional zero-shot learning methods generally perform worse than the recent methods designed for zero-shot node classification. Maybe the reason is that traditional zero-shot learning methods make it difficult to capture the characteristics of complex graph-structured data, while DPGN and DBiGCN excel at exploring the relational information between nodes in graph datasets.

- For all datasets, our proposed GraphCEN outperforms strong baselines DPGN and DBiGCN by a significant margin, which suggests that our approach is more effective in capturing dependencies between nodes and nodes, nodes and classes. The two-level contrastive learning can also inherently learn the representations with generalization ability from the joint information matrix, which is more beneficial for the knowledge transfer from seen classes to unseen classes.

- To validate the scalability of our proposed method, we conduct experiments on the large-scale dataset ogbn-arxiv. We compare it with the two latest baselines DPGN and DBiGCN, and from

Table 2: The overall performance (%) on three benchmark datasets for zero-shot node classification. The best results are shown in boldface and the second-best is underlined.

| | | Cora | Citeseer | C-M10M |
|---|---|---|---|---|
| **Class Split I** | RandomGuess | 25.35 | 24.86 | 33.21 |
| | DAP | 26.56 | 34.01 | 38.71 |
| | DAP(CNN) | 27.80 | 30.45 | 32.97 |
| | ESZSL | 27.35 | 30.32 | 37.00 |
| | ZS-GCN | 25.73 | 28.62 | 37.89 |
| | ZS-GCN(CNN) | 16.01 | 21.18 | 36.44 |
| | WDVSc | 30.62 | 23.46 | 38.12 |
| | Hyperbolic-ZSL | 26.36 | 34.18 | 35.80 |
| | DGPN | 33.76 | 37.74 | 41.93 |
| | DBiGCN | 45.08 | 38.57 | 41.11 |
| | GraphCEN (Ours) | **48.43** | **40.77** | **44.17** |
| | Improve ↑ | +7.43% | +5.70% | +5.34% |
| **Class Split II** | RandomGuess | 32.69 | 50.48 | 49.73 |
| | DAP | 30.22 | 53.30 | 46.79 |
| | DAP(CNN) | 29.83 | 50.07 | 46.29 |
| | ESZSL | 38.82 | 55.32 | 56.07 |
| | ZS-GCN | 29.53 | 52.22 | 55.28 |
| | ZS-GCN(CNN) | 33.20 | 49.27 | 51.37 |
| | WDVSc | 34.13 | 52.70 | 46.26 |
| | Hyperbolic-ZSL | 37.02 | 46.27 | 55.07 |
| | DGPN | 48.31 | 58.86 | 61.68 |
| | DBiGCN | 46.95 | 58.37 | 66.12 |
| | GraphCEN (Ours) | **50.61** | **60.47** | **70.83** |
| | Improve ↑ | +4.76% | +2.74% | +7.12% |

Table 3: The comparison (%) of DGPN, DBiGCN, and our proposed GraphCEN on the large-scale ogbn-arxiv dataset for zero-shot node classification.

| | DGPN | DBiGCN | GraphCEN (Ours) |
|---|---|---|---|
| **Class Split I** | 22.37 | 21.40 | **23.96** |
| **Class Split II** | 21.95 | 25.92 | **28.36** |

Table 3, it can be seen that our method GraphCEN still outperforms others on the large-scale dataset, particularly in the class split II, highlighting the strong discriminative ability and better generalization achieved through our dual-contrastive learning. This fully demonstrates that our method GraphCEN learns more powerful representations and exhibits better scalability.

### 4.3 Ablation Study

To provide further insights into the GraphCEN, we conduct the ablation study to evaluate the effectiveness of the three main components, i.e., node-level contrast (N-level) and class-level contrast (C-level + Entropy). Next, we explore six different variants designed as follows:

- $M_1$: Our base model, which trains a GNN solely on labeled data in a fully supervised manner with training objective $\mathcal{L}_{\text{CE}}$;

Table 4: Analysis of ablation study. N-level, C-level and Entropy represent that node-level contrastive loss, class-level contrastive loss or the entropy function is applied, respectively.

| | Contrastive Item | | | Accuracy (%) | | |
|---|---|---|---|---|---|---|
| | N-level | C-level | Entropy | Cora | Citeseer | C-M10M |
| $M_1$ | | | | 45.47 | 38.18 | 33.38 |
| $M_2$ | | $\checkmark$ | | 45.94 | 38.71 | 33.52 |
| $M_3$ | | $\checkmark$ | $\checkmark$ | 46.14 | 38.81 | 33.47 |
| $M_4$ | $\checkmark$ | | | 47.96 | 39.25 | 36.85 |
| $M_5$ | $\checkmark$ | $\checkmark$ | | 48.09 | 39.60 | 43.71 |
| $M_6$ | $\checkmark$ | $\checkmark$ | $\checkmark$ | **48.43** | **40.77** | **44.17** |

- $M_2$: It is a variant with only class-level contrast except for the entropy function where we optimize the GNN with both training objectives $\mathcal{L}_{\mathrm{CE}}$ and $\mathcal{L}_{\mathrm{CCL}} + H(\mathbf{Z})$;

- $M_3$: It is a variant with only class-level contrast where we optimize the GNN with both training objectives $\mathcal{L}_{\mathrm{CE}}$ and $\mathcal{L}_{\mathrm{CCL}}$;

- $M_4$: It is a variant with only node-level contrast where we optimize the GNN with both training objectives $\mathcal{L}_{\mathrm{CE}}$ and $\mathcal{L}_{\mathrm{NCL}}$;

- $M_5$: It is a variant with both node-level and class-level contrasts except for the entropy function where we optimize the GNN with both training objectives $\mathcal{L}_{\mathrm{CE}}$ and $\mathcal{L}_{\mathrm{NCL}} + H(\mathbf{Z})$;

- $M_6$: Our full model, which combines both node-level and class-level contrasts with both training objectives $\mathcal{L}_{\mathrm{CE}}$ and $\mathcal{L}_{\mathrm{CL}}$.

The results of the above six variants are summarized in Table 4. The performance of models $M_3$ and $M_4$ shows the superiority over model $M_1$ on all datasets, indicating that the introduction of both class-level contrast and node-level contrast alone can indeed improve overall performance. For all datasets, the performance of model $M_6$ which incorporates both class-level and node-level contrasts is superior to the other variants. We can infer that either node-level or class-level contrast is indispensable, which implies that the combination of both levels leads to more discriminative and generalizable node embeddings for effective class assignments, thus being beneficial for zero-shot node classification. Additionally, when comparing $M_2$ with $M_3$ and $M_5$ with $M_6$, we observe that the increase in Entropy is relatively small, except when contrasting $M_5$ and $M_6$ on the Citeseer. The reason might be that class-level contrast already ensures the learning of discriminative class representations, implicitly preventing collapse to trivial solutions. However, considering that Entropy can further guarantee the model's optimization for different datasets, playing an explicit protective role, we choose to retain this term, and the experimental gains also demonstrate the necessity of Entropy.

## 4.4 Sensitivity Analysis

In this section, we look into the sensitivity of parameters: contrastive weight $\beta$, contrastive temperature $\tau$.

**Effect of Contrastive Weight.** We first examine the impact of the contrastive weight on two datasets Cora and Citeseer by varying $\beta$ from 0.0 to 10.0. As depicted in Figure 2(a), the performance on Cora improves gradually when $\beta$ increases from 0.0 to 0.1 and remains stable from 0.2 to 2.0. We attribute this improvement to the introduction of both class- and node-level contrastive learning. Nevertheless, excessive contrastive weight on Cora may harm performance due to the neglect of the supervised loss, biasing the correct direction of gradient learning. Additionally, the performance on Citeseer exhibits a similar trend of improvement when $\beta$ increases from 0.0 to 0.2 However, our approach is not sensitive to parameter variations from 2.0 to 5.0 on Citeseer.

**Effect of Contrastive Temperature.** We then conduct experiments on datasets Cora and Citeseer to evaluate the sensitivity to contrastive temperature $\tau$. As shown in Figure 2(b), the accuracy on both datasets

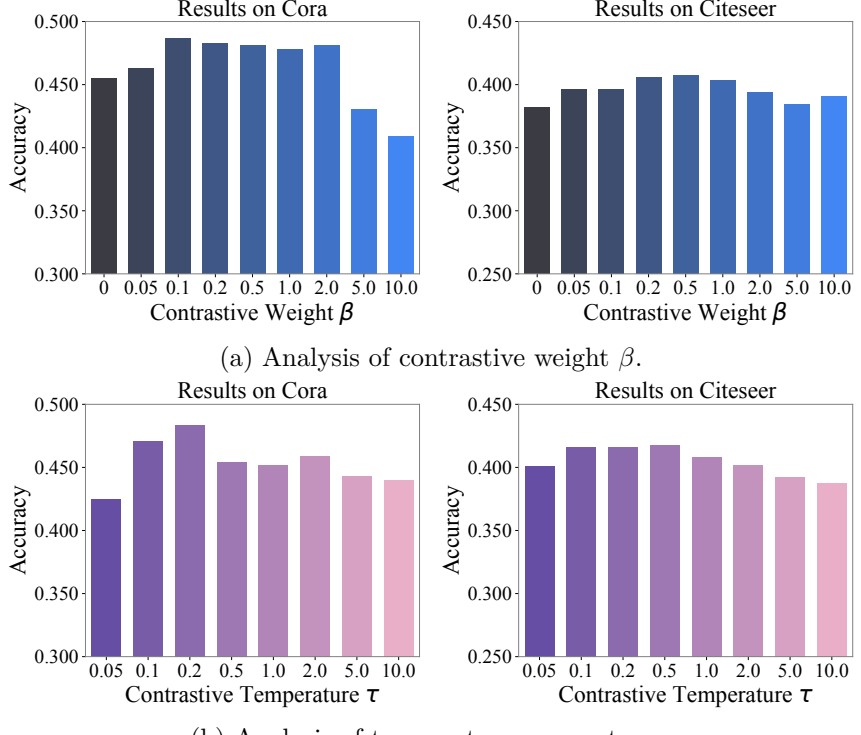

(a) Analysis of contrastive weight $\beta$.

(b) Analysis of temperature parameter $\tau$.

Figure 2: Analysis of parameter $\beta$ and $\tau$ in contrastive learning.

Table 5: Accuracy (%) of zero-shot node classification w.r.t. different CSDs on Cora.

| | | Cora | | |
| | | TEXT-CSDs | LABEL-CSDs | Decline rate |
|---|---|---|---|---|
| | DAP | 26.56 | 25.34 | -4.59% |
| | ESZSL | 27.35 | 25.79 | -5.70% |
| | ZS-GCN | 25.73 | 23.73 | -7.77% |
| Class **Split I** | WDVSc | 30.62 | 18.73 | -38.83% |
| | Hyperbolic-ZSL | 26.36 | 25.47 | -3.38% |
| | DGPN | 33.76 | 32.69 | -3.17% |
| | DBiGCN | 45.08 | 32.89 | -27.04% |
| | GraphCEN (Ours) | **48.43** | **39.63** | -18.07% |

reaches its peak when $\tau$ is set to 0.2, and gradually decreases as $\tau$ continues to increase. This is because a higher temperature will result in a softer distribution, where all classes are more likely, while a lower temperature will result in a sharper distribution, where some classes are much more likely than others. In other words, a higher temperature would make the model less confident in its predictions, which would result in lower accuracy.

## 4.5 Discussion on Different CSDs

Since there are two kinds of CSDs, we can construct the class affinity graph defined in Eq. 2 based on them. As the two CSDs provide different semantic information for the class affinity graph $G_A$, the model performance on zero-shot node classification, which is highly dependent on the label semantics, would be significantly affected by the chosen CSDs. Specifically, the accuracy of zero-shot node classification w.r.t. the used CSD type is shown in Table 5, 6 and 7. It can be observed that the performance with TEXT-CSDs is

Table 6: Accuracy (%) of zero-shot node classification w.r.t. different CSDs on Citeseer.

| | | Citeseer | | |
| | | TEXT-CSDs | LABEL-CSDs | Decline rate |
|---|---|---|---|---|
| | DAP | 34.01 | 30.01 | -11.76% |
| | ESZSL | 30.32 | 28.52 | -5.94% |
| | ZS-GCN | 28.62 | 26.11 | -8.77% |
| Class Split I | WDVSc | 23.46 | 19.70 | -16.02% |
| | Hyperbolic-ZSL | 34.18 | 21.04 | -38.44% |
| | DGPN | 37.74 | 31.05 | -17.73% |
| | DBiGCN | 38.57 | 34.18 | -11.38% |
| | GraphCEN (Ours) | **40.77** | **38.45** | -5.69% |

Table 7: Accuracy (%) of zero-shot node classification w.r.t. different CSDs on C-M10M.

| | | C-M10M | | |
| | | TEXT-CSDs | LABEL-CSDs | Decline rate |
|---|---|---|---|---|
| | DAP | 38.71 | 32.67 | -15.60% |
| | ESZSL | 37.00 | 35.02 | -5.35% |
| | ZS-GCN | 37.89 | 33.32 | -12.06% |
| Class Split I | WDVSc | 38.12 | 30.82 | -19.15% |
| | Hyperbolic-ZSL | 35.80 | 34.49 | -3.66% |
| | DGPN | 41.93 | 35.12 | -16.24% |
| | DBiGCN | 41.11 | 37.54 | -8.68% |
| | GraphCEN (Ours) | **44.17** | **38.68** | -12.43% |

higher than with LABEL-CSDs, which is consistent with previous researches (Wang et al., 2021b; Yue et al., 2022). The result shows the expressiveness of natural language when it comes to representing label-label relationships. Among the listed models, our GraphCEN achieves the best performance no matter which kind of CSDs are used, which further demonstrates the robustness of the proposed approach via incorporating the two-level contrastive learning.

## 4.6 Analysis of Graph Augmentation

To explore the effect of different graph augmentations, we compare the performance of three strategies, i.e., original graph, edge dropping, and graph diffusion. The horizontal axis denotes the augmentation strategy applied to generate $\mathbf{H}^a$, while the vertical axis represents the strategy for generating $\mathbf{H}^b$. The results on two datasets are reported in Figure 3. It can be observed that different augmentation strategies exhibit vital effects. GraphCEN with {ED + GD} obtains the best performance, whereas {OG + OG} (without augmentations) yields the worst results. Maybe the reason is that different augmentations can maximize the diversity of graph data while preserving semantics, which better promotes consistency in contrastive learning and enhances generalization for zero-shot node classification. Moreover, we find that the effect of {GD + GD} is worse than that of {GD + OG}. The possible reason is the generation of graph diffusion is not random, resulting in the same two augmented views, which hinders the diversity of the augmentation process, while {ED + ED} with randomness will lead to different augmented views, and thus performs better than that of graph diffusion. Besides, augmentation with {ED + OG} is worse than {GD + OG}. The explanation for this is that graph diffusion is likely to capture the global information instead of exploring from a local view like edge dropping, which results in inadequate semantic information exploration. Overall, the aforementioned experiments demonstrate the importance of graph augmentation in contrastive learning. Encouraging consistency with diverse data augmentations maximizes the model's ability to learn robust semantic representations, enhances its generalization capabilities, and proves beneficial for our zero-shot node classification task. If only the original graphs are used without augmentation, the model may become highly

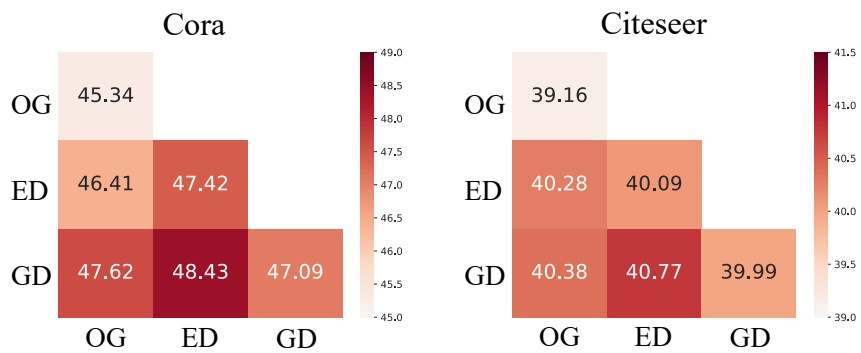

Figure 3: Analysis of different graph augmentations on datasets Cora and Citeseer. OG, ED, and GD correspond to original graph, edge dropping, and graph diffusion, respectively. Here the horizontal axis denotes the augmentation strategy applied to generate $\mathbf{H}^a$, while the vertical axis represents the strategy for generating $\mathbf{H}^b$.

sensitive to data perturbations or variations, hampering its ability to learn discriminative representations and consequently affecting the model's performance.

## 5    Related Work

**Node Classification.** As one of the most fundamental problems in relational data modeling, many of the real-life applications can be boiled down to the problem of semi-supervised node classification. It aims to predict the unlabeled nodes with only a few labeled nodes on the graph. This problem has been extensively investigated with graph neural networks (GNNs) (Ju et al., 2023a; Nagarajan & Raghunathan, 2023; Ma et al., 2023; Prieto et al., 2023; Zhong et al., 2022; Chen et al., 2023; Ju et al., 2023d; Demirel et al., 2021) and can generalize well to downstream domains such as social analysis (Bhagat et al., 2011; Yuan et al., 2023) and bioinformatics (Yuan et al., 2021; Gasteiger et al., 2022). Benefiting from the powerful capability to incorporate topological structure and associated features, GNNs have shown outstanding performance in learning effective node embeddings. Nevertheless, these methods have the inability to handle zero-shot node classification, while our proposed GraphCEN models the dependencies between nodes and classes to transfer knowledge from the seen classes to the unseen classes.

**Zero-shot Learning.** Our work essentially belongs to zero-shot learning (ZSL), which has been extensively studied in computer vision. Inspired by human cognitive competence, this topic has recently intrigued vast interest, with the capability of recognizing new classes during learning by exploiting the intrinsic semantic relationship between seen and unseen classes, with the guidance of some auxiliary information. Existing ZSL methods can be divided into two main categories: (i) embedding-based methods (Akata et al., 2015; Frome et al., 2013; Lampert et al., 2013), and (ii) generative-based methods (Xian et al., 2018; 2019; Schonfeld et al., 2019; Guan et al., 2020). For the first category, it aims to learn an embedding mapping function from visual space to semantic space. For the second category, the basic idea is to apply a generative model to synthesize features of unseen classes. However, these methods are not tailored to zero-shot node classification, and fail to be capable of processing complex data structures, such as graph domains. Recently, DGPN (Wang et al., 2021b) and DBiGCN (Yue et al., 2022) have been proposed for zero-shot node classification. DGPN designs the acquisition of high-quality class semantic descriptions (CSDs) and follows the principles of locality and compositionality, while DBiGCN proposes two dual GCNs with two opposite directions and introduces a label consistency loss to mutually enhance. Different from them, our proposed model GraphCEN starts from the relationship between nodes and classes, and utilizes contrastive learning to mine the intrinsic connections between semantics, thus promoting knowledge transfer more effectively.

**Contrastive Learning.** Another category of related work is contrastive learning (CL), which has recently achieved state-of-the-art performance in the direction of self-supervised learning, arousing extensive attention from researchers. The core idea of CL is based on the task of instance discrimination (Wu et al., 2018) and

introducing a point contrastive loss that encourages the matched positive point pairs to be similar in the embedding space while pushing away the negative pairs (Hadsell et al., 2006). Recently, there are many CL approaches proposed to capture the discriminative representations (Chen et al., 2020a; He et al., 2020; Chen et al., 2020b; Grill et al., 2020). For example, SimCLR (Chen et al., 2020a) trains an encoder by adopting multiple data augmentations and a learnable nonlinear transformation to pull the feature embeddings from the same images. MoCo (He et al., 2020) develops a dynamic dictionary for CL via a moving-averaged encoder. SupCon (Khosla et al., 2020) extends the self-supervised contrastive approach to the fully supervised setting, which allows for effective leverage of the label information. Additionally, many recent methods are extending CL to graph domains (You et al., 2020; Ju et al., 2023c; Yi et al., 2023; Luo et al., 2022; Ju et al., 2023b). Compared with existing CL methods, our work goes further and explores the node- and class-level CL simultaneously for zero-shot node classification on graphs.

## 6 Conclusion

In this paper, we develop a novel framework GraphCEN for zero-shot node classification, which explicitly models the dependencies between nodes and classes to transfer knowledge from seen classes to unseen classes. We first construct a class affinity graph to capture similar category semantics, and then integrate the class affinity graph and node features into the joint information matrix via GNNs. Further, we introduce node- and class-level contrastive learning based on the information matrix for effective class assignments. Experimental results demonstrate that our proposed GraphCEN consistently outperforms existing state-of-the-art methods.

As our future work, there are several aspects of the proposed model that deserve further investigation: (i) leveraging the neighbor information of constructed class affinity graph to conduct better contrastive learning; (2) adopting the unsupervised pre-training techniques to sufficiently explore inherent graph semantics for better knowledge transfer; (iii) extending our framework to more challenging settings such as generalized zero-shot learning or open-world graph learning.

### Acknowledgments

This paper is partially supported by the National Natural Science Foundation of China with Grant (NSFC Grant No. 62106008, No. 62276002 and 62306014) as well as the China Postdoctoral Science Foundation with Grant No. 2023M730057.

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
