# OpenReview forum: "Zero-shot Node Classification with Graph Contrastive Embedding Network"
_TMLR — Accepted by TMLR_

### Review · Reviewer_rL7A · 2023-07-02

**Summary Of Contributions:**

 First, thank you to the authors for an interesting piece of work. I enjoyed reading the ideas, results, and various connections it all points to.

**Brief summary:**

The paper addresses the problem of zero-shot node classification in graphs, which involves predicting the classes not seen during training. The authors propose a method that combines contrastive learning and supervised learning, using the unsupervised component to enforce a consistent structure/relation between the seen and unseen classes, thereby working towards that stated goal of generalizing to unseen classes.


**Audience:**

Yes

**Claims And Evidence:**

Yes

**Requested Changes:**

**Questions:**

- What is the effect of $A^s$? This is included as part of the model architecture but is orthogonal to the contrastive learning loss proposed. How does your method perform without this $A^s$? If the method is only beating baselines due to this design component that could equally be added to other baseline methods then that’s a concern. A response to this point is an absolute must, as right now I am concerned that the experimental setting is unfair on the baselines and need reassurance that this is not the case. Or is the use of $A^s$ in this way a novel contribution? If this is so, an explanation of how this paper’s use of $A^s$ differs from the original Wang et al. 2021 would be appreciated.

- Why have you added the entropy $H(Z)$ to your contrastive loss “to prevent collapsing into trivial outputs of the same class”? I would be grateful for some clarification of this component, since to my eyes it seems superfluous - the negative samples are there precisely to avoid this collapse, so why the need for this term? Can this term also be removed from method safely without hurting performance?

**Strengths And Weaknesses:**

**Discussion:**

The authors suggest a contrastive learning method that acts on an embedding matrix $Z$ of dimensions $n \times c$ where $n$ is the number of nodes, and $c$, the number of classes (including unseen ones). The method combines _two_ contrastive learning objectives - one acting along the rows of $Z$ (node-wise_ and the other along the columns of $Z$ (class-wise).

The node-wise contrastive learning simply considers two views of the original graph—-generated in the usual kind of way by dropping edges at random and applying a diffusion operator to the adjacency matrix. The class-wise loss acts similarly but on the columns. Column $i$ of the embedding of both views are trained to be the same, and all other pairs of columns are treated as negative pairs. One way to think of this approach is as normal (graph) contrastive learning, but instead of arbitrary node embedding dimension, the dimension _has to be $c$_, the number of classes, and the column-wise loss forces each dimension of vector to be decorrelated from the others (quite similar to Barlow Twins in this way).

Finally, to actually force the trained-to-be-decorrelated embedding dimensions to actually correspond to particular classes, the authors introduce a third loss - a standard cross-entropy loss on the classes that are seen during training (recall, the setting assumes only some of the classes are seen in training).

Each of the three losses used in the method does seem to address different desired properties of the model and broadly make sense in combination.

---

**Conclusion:**

The problem setting is interesting enough, and the method broadly makes sense (see questions for concerns) and seems to produce decent results. I expect that the TMLR audience would find this work interesting. A caveat is that I am not abreast of the state-of-the-art in this problem setting, so am unable to assess the similarities to prior work.

However, I have a couple of concerns that relate to critical parts of the framework. I have laid out these concerns under the “questions” section.   Overall I am likely trending toward accepting this paper. However to clearly register my concern with these questions, and the need for a clear answer to them, I have currently assigned a weak reject score. This will most likely rise after hearing the author’s response.


---

**Writing:**
In terms of writing clarity, I would like to be firm. I would like the authors to know that the spirit of this message is to improve the accessibility of their work to readers, and does not form a part of my assessment of the quality of the work, which determines my advocation for acceptance or rejection.

The writing needs considerable revision. Primarily I am concerned that sections 2 and 3 (setup and method) are organized poorly, making the ideas harder to digest than need be. Specifically, the information is either presented in an incomplete, or non-linear form. Some examples of what I mean by this:

**Incomplete:**
- Diffusion Equation 1 - you neglect to actually mention what the diffusion matrix $A’$ is used for. I am assuming that the adjacency matrix is simply updated to be this new diffused matrix. But if this is actually the case you should explicitly say so and do not assume the reader’s familiarity with what people use this diffusion for in graph contrastive learning.

- The class affinity graph (sec 3.2) is never properly defined. Specifically, the definition of the neighborhood N(s) of a class is never defined, and a paper from 2021 is simply cited and readers are directed to read it. What makes this worse is that this 2021 paper does not use the words “class affinity graph” at all, and does not use the notation N(s) at all, making the reader have to dig through their nomenclature to find the relevant definition.

**Non-linear**
- Section 3.2 again. The “class affinity graph” is (partially) defined, but the reader does not yet know what it is for. It is only in the next section that its use in this context is revealed. This form of writing is common in mathematics, but machine learning audiences likely would like to know the big-picture of why this math is being introduced before it is introduced.

The authors also are inconsistent in notation. $H$ and $Z$ denote the same matrix in different places (one when doing row-wise, and the other when doing column-wise loss). I see the reasons for a switch-up, but make the transition clearer. Related to this, there are often vague or unclear phrases used, including at key moments when the authors are trying to establish the motivation for their work.

For example, “Inability to explicitly model the dependencies between nodes and classes” (Key limitation ii). The authors should try to make this much more specific since I don’t know what this means in its current form. I suspect the idea is that the model should be using label information _as part of its layer-wise updates_ - i.e., do not just use class information on output embedding vectors via cross-entropy loss etc., but actually use the labels to determine information propagation through the model. Finally, although this sounds a sensible enough sentiment, it seems generic and not directly related to the question of zero-shot node classification (see questions for more on this).

---

### Review · Reviewer_GSsH · 2023-07-14

**Summary Of Contributions:**

In this work the authors present a GraphCEN, a model for zero-shot node classification whose architecture enables the use of contrastive training objectives on both nodes and classes. The main novel contribution is the incorporation of class affinity information, which is captured as an adjacency matrix with values related to the dot product of feature vectors for each class. This matrix multiplies the output of a graph neural network (which aggregates node embeddings) to create what the authors refer to as the "joint information matrix", presumably because the rows of this matrix (which provide a distribution over classes) can be viewed as node embeddings, while the columns (which provide a distribution over nodes) can be viewed as class embeddings. They leverage this interpretation to provide a contrastive learning objective, wherein slightly different joint information matrices are produced as a result of augmenting the underlying graph, and corresponding rows and/or columns are encouraged to be more similar to each other than others.

The authors run zero-shot node classification experiments on three datasets, demonstrating state-of-the-art performance on all three. They also include comprehensive ablations considering various aspects of the contrastive loss.

**Audience:**

Yes

**Claims And Evidence:**

Yes

**Requested Changes:**


1. Overall, there are a number of issues with the mathematical formulation, notation, and rigor in the paper. For example: when defining the class affinity graph (around equation (2)) the vertices are defined as $\mathcal V_G = \{v_1, \ldots, v_{|\mathcal C|}\}$, but this notation was already used to describe the nodes of the original graph $G$. Thus, a strict interpretation of what was described would amount to the class affinity graph using nodes from the original graph as opposed to classes, which is what was presumably intended. One way to fix this would be to simply define $\mathcal V_G = \mathcal C$ (after all, that's the point of abstracting graphs away from the actual nodes they connect) however since we don't actually use the graph structure and rather simply use the adjacency matrix $\mathbf A^s$ (which, incidentally, does not have entities in $\{0,1\}$ and therefore the graph should be qualified as a *weighted* graph) I think it would be far simpler to avoid the whole auxiliary graph definition entirely and simply state that you create a similarity matrix as described in equation (2). Furthermore, the superscript $s$ was already used in describing the dimension of the feature vectors for the classes, thus making $\mathbf A^s$ look like a matrix $\mathbf A$ raised to an integer power $s$. Instead, you presumably intended this superscript on $\mathbf A$ to indicate a dependency on the feature vectors themselves, e.g. $\mathbf A ^{\mathbf S}$. This is fairly representative of the issues with mathematical notation throughout the paper, which leads to significant confusion in various points.
2. Space does not seem to be at a premium here, and it would be worthwhile to expand on the technical details of the model so that the reader would actually be capable of reproducing the work directly. For example:
    1. The paper would become more self-contained if a definition of the neighbors $\mathcal N(s)$ was included near equation (2).
    2. In section 3.4.1 it is unclear to me what is meant by "After encoding from our well-designed GNN in Section 3.3 followed by a node head (i.e., multi-layer perceptron, MLP), we can obtain the augmented joint information matrices $\mathbf H^a$ and $\mathbf H^b$." Are you saying that $\mathbf H^a = W_2\operatorname{ReLU}(W_1 \operatorname{ReLU}(\hat{\mathbf A^a} \mathbf X \mathbf W) \mathbf A^s)$ and  $\mathbf H^b = W_2\operatorname{ReLU}(W_1 \operatorname{ReLU}(\hat{\mathbf A^b} \mathbf X \mathbf W) \mathbf A^s)$, where $\hat{\mathbf A^x} = R(\operatorname{Aug}_x(\mathbf A))$, $R(\mathbf A) = \widetilde {\mathbf D}^{-\frac 1 2} \widetilde{\mathbf A} \mathbf{D}^{-\frac 1 2}$ for some adjacency matrix $\mathbf A$, where $\widetilde{\mathbf A} = \mathbf A + \mathbf I$ and $\widetilde{\mathbf D}$ is it's degree matrix?
    3. After equation (8) it would be useful to define exactly the function $H(\cdot)$ as opposed to just stating it is the entropy function, because $\mathbf Z$ is not a random variable and so it is not clear to me how this is defined.
4. In Section 4.1 some details of the datasets are provide, including the fact that nodes represent different publications and the edges represent citation relationships, however there is no information about what the classes are. This seems rather crucial, given the main goal of the paper.
5. In Section 4.6 you analyze the differences between various choices of graph augmentation, and present results in Figure 3, however I could not find any information about what the horizontal or vertical axes represent. I assume one is for the augmentation applied to the nodes, and the other is the augmentation applied to the classes when performing the contrastive loss. When doing so, one could either (a) perform the ablation on the original graph and then create the embeddings, or (b) apply the ablation only for the part of the model which utilizes the adjacency matrix (eg. if the method "ED" is used for the nodes, it could be used to generate a new matrix $\mathbf A$ which is then used for both $\hat {\mathbf A}$ and $\mathbf A^s$ when creating the augmented node embeddings, or it could be used for just $\mathbf A$). In writing this I recognize that the answer may also be "neither", particularly since the resulting Figure 3 does not have the elements in the upper-right (presumably because they would be symmetric, which is not the case with my two proposed interpretations). If I missed a description of this I apologize, however I think the difficulty in understanding here further highlights the benefit of using clear notation and writing explicitly the function which ultimately computes the embeddings in these ablation settings as well. Perhaps, instead of either my interpretations, this analysis considers the augmented graphs used to produce $\mathbf H^a$ and $\mathbf H^b$, but in this case it is very unclear to me why using two augmented graphs would ever be better than using the original graph in some capacity.

**Strengths And Weaknesses:**

At a high level the work in this paper clearly meets the requirements for acceptance criteria in TMLR. The model is supported by ample empirical evidence and ablation studies, and people in the community will certainly find the results interesting.

The writing of the paper leaves much to be desired, however. There is a long list of typos which are substantial enough to make some parts of the paper hard to parse. More problematic, however, is that certain parts of the model itself were not presented completely or clearly, to the point where I have no confidence that I would be able to reproduce the author's proposed architecture from reading the paper alone. In addition, to the extent that I understood it, certain parts of the proposed model seem suboptimal, and some explanations in the ablation studies are confusing. I will go into more detail in the following section.

---

### Review · Reviewer_CMeh · 2023-07-17

**Summary Of Contributions:**

This paper studies zero-shot node classification problem, where each class also comes with a semantic description vector. The author proposed GraphCEN, which first forms the node-to-label joint information matrix, and then performs  node and class level contrastive learning based on the rows/columns of the join information matrix. Empirically, their proposed GraphCEN outperforms other state-of-the-art zero-shot classification methods. The author also conducted several ablation studies to justify the effectiveness of each components in GrpahCEN.

**Audience:**

Yes

**Broader Impact Concerns:**

To my knowledge, there's no concerns regarding the ethical implications of the work.

**Claims And Evidence:**

Yes

**Requested Changes:**

**Technical Details**
- Q1: In Algorithm 1, why constructing the class affinity graph `G_A` inside the while loop? If `G_A` is just the kNN graph from the class semantic description vectors `s_k`, then `G_A` is not subject to change right?
- Q2: It is not clear how the model parameter `\theta` is used in Sec 3.4.1 and Sec 3.4.2. Is it the MLP parameters to produce $H^a H^b, Z^a, Z^b$ from $H$ in Equation (4) ?
- Q3: For time complexity analysis in Section 3.6, why $d^{'}$ is not used in the BigO notation of computing the joint information matrix? To compute $\text{ReLU}(A^{hat}XW)$, we need to first multiply the node feature matrix $X \in \mathbb{R}^{N \times d}$ with the weight matrix $\mathbb{R}^{d \times d'}$, which takes $O(N \cdot d \cdot d^{'})$?

**Main Experiment Results**
- Q1: There are some discrepancy between the numbers of `DGPN` in Table 2 of this submission versus the original numbers in Table 3 of DGPN paper. In this submission, the Class Split II accuracy of `DGPN` are 59.38 and 60.12 for Citeseer/C-M10M, respectively. However, in the original paper, the numbers are 61.90 and 62.46 for Citeseer/C-M10M, respectively. Any explanation?
- Q2: There are **large** discrepancy between the numbers of `DBiGCN` in Table 2 of this submission versus the original numbers in Table 3 of DBiGCN paper. In this submission, the Class Split II accuracy of `DBiGCN` are 58.29 and 66.23 for Citeseer/C-M10M, respectively. However, in the original paper, the numbers are 60.11 and 71.46 for Citeseer/C-M10M, respectively. Any explaination?

**Ablation Studies**
- Q1: How important is the entropy regularization term in Equation(8)? Any ablation results on that?
- Q2: How scalable is the proposed method? Any results on larger scale datasets (e.g., million numbers of nodes)?

**Strengths And Weaknesses:**

**Strengths**
- The writing is clear and the paper is easy to follow
- The proposed method outperforms other competitive baselines

**Weaknesses**
- The scalability to large number of nodes and classes (e.g., millions or more) seems challenging
- There discrepancy in the accuracy metrics for two major baselines (DGPN/DBiGCN). The numbers presented in this submission are different from the numbers presented in the DGPN and DBiGCN papers.

---

### Decision · Action_Editors · 2023-09-10

**Recommendation:** Accept with minor revision

**Comment:**

The paper aims to improve zero-shot node classifications in graphs. In this paper, the authors present a framework which first constructs an affinity graph to model the relations between the classes and then a node- and class-level contrastive learning are used to jointly learn node embeddings and class assignments in an end-to-end manner. However presentation and writing of the paper leaves more to be desired. We thank the authors and reviewers to actively engage in discussion for making the paper better. During discussion many of the reviewer concerns/questions were resolved like the one about discrepancy in the accuracy metrics or role of $A^*$. However some concerns remain about the presentation and writing. Nevertheless, the proposed method and technique are correct, experimental result demonstrate robustness to quantization in one way, and the approach will be of interest to the community, hence I propose to accept the paper with some minor modifications. Please improve the presentation and writing as pointed out by the reviewers. Note that errors pointed out by were not exhaustive, and there may still be other areas for improving the writing.

**Audience:**

Yes, both graph learning and few-shot/zero-shot learning community will be interested in this paper.

**Claims And Evidence:**

Yes